# Third COVID-19 vaccine dose boosts neutralizing antibodies in poor responders

Douglas F. Lake[1,5 ✉], Alexa J. Roeder[1,5], Maria J. Gonzalez-Moa[2], Megan Koehler[1], Erin Kaleta[3], Paniz Jasbi[4], John Vanderhoof[1], Davis McKechnie[1], Jack Forman[1], Baylee A. Edwards[1], Alim Seit-Nebi[2] & Sergei Svarovsky[2 ✉]

## Abstract

**Background** While evaluating COVID-19 vaccine responses using a rapid neutralizing antibody (NAb) test, we observed that 25% of mRNA vaccine recipients did not neutralize >50%. We termed this group "vaccine poor responders" (VPRs). The objective of this study was to determine if individuals who neutralized <50% would remain VPRs, or if a third dose would elicit high levels of NAbs.

**Methods** 269 healthy individuals ranging in age from 19 to 80 (Average age = 51; 165 females and 104 males) who received either BNT162b2 (Pfizer) or mRNA-1273 (Moderna) vaccines were evaluated. NAb levels were measured: (i) 2–4 weeks after a second vaccine dose, (ii) 2–4 months after the second dose, (iii) within 1–2 weeks prior to a third dose and (iv) 2–4 weeks after a third mRNA vaccine dose.

**Results** Analysis of vaccine recipients reveals that 25% did not neutralize above 50% (Median neutralization = 21%, titers <1:80) within a month after their second dose. Twenty-three of these VPRs obtained a third dose of either BNT162b2 or mRNA-1273 vaccine 1–8 months (average = 5 months) after their second dose. Within a month after their third dose, VPRs show an average 5.4-fold increase in NAb levels (range: 46–99%).

**Conclusions** The results suggest that VPRs are not permanently poor responders; they can generate high NAb levels with an additional vaccine dose. Although it is not known what levels of NAbs protect from infection or disease, those in high-risk professions may wish to keep peripheral NAb levels high, limiting infection, and potential transmission.

## Plain language summary

Neutralizing antibodies are proteins used by the immune system to respond to viruses and other infectious agents. Vaccination against COVID-19 induces production of neutralizing antibodies that stop virus from infecting cells. We measured levels of neutralizing antibodies in a drop of blood after 2 doses of vaccines distributed by Pfizer and BioNTech or Moderna (COMIRNATY and Spikevax). Twenty-five percent of vaccine recipients did not make high levels of neutralizing antibodies. After receiving a third dose of vaccine, most of these vaccine recipients made high levels of neutralizing antibodies. Our data suggest a third dose is important for vaccine recipients that did not generate high neutralizing antibody levels after 2 doses of vaccine and thus might be an important component of a successful vaccination strategy.

[1] Arizona State University, School of Life Sciences, Tempe, AZ, USA. [2] AXIM Biotechnologies, San Diego, CA, USA. [3] Mayo Clinic Arizona, Department of Laboratory Medicine and Pathology, Scottsdale, AZ, USA. [4] Arizona State University, College of Health Solutions, Phoenix, AZ, USA. [5] These authors contributed equally: Douglas F. Lake, Alexa J. Roeder. ✉email: douglas.lake@asu.edu; ssvarovsky@sapphirebio.com

COVID-19 mRNA vaccines prevent serious clinical disease requiring hospitalization in ~95% of vaccine recipients. This suggests that 5% of vaccinated individuals remain susceptible to potentially severe disease[1,2]. If 300 million people receive two doses of the COVID-19 mRNA vaccines, then approximately 15 million people may not be fully protected. Although T cells are important in anti-viral immunity, their activity is difficult to rapidly evaluate at scale. Furthermore, if T cells are engaged, the host is already infected. After natural infection with SARS-CoV-2 or vaccination against COVID-19, anti-viral antibodies are generated by the host. Antibodies of primary importance are neutralizing antibodies (NAbs) because they prevent infection. However, non-neutralizing antibodies may also play an important role in the host's humoral response[3,4]. NAbs block the spike protein on SARS-CoV-2 from binding to the host cell receptor, angiotensin converting enzyme 2 (ACE2). In particular, the portion of the spike protein that binds to ACE2 is the receptor binding domain (RBD)[5,6] and there have been many reports of natural, vaccine-induced[7–10] and therapeutic antibodies[11] that neutralize the virus by binding to the RBD.

After 2 doses of either BNT162b2 or mRNA-1273, antibodies to spike protein and neutralizing antibodies have been quantified in vaccine recipients[1,2,12,13]. Durability of those responses has been reported[14,15]. Although NAb titers as a correlate of protection remain undefined and are complicated by evolving variants, titers that provide protection from disease likely differ from titers that prevent infection. Both are also largely dependent on the dominant variant in circulation. Even when vaccinated, immunosuppressed individuals are at increased risk of infection and disease if exposed[16,17]. As such, caregivers for high-risk individuals may want to measure or monitor their NAb levels after their last vaccination or natural infection, to lessen the possibility of asymptomatic infection which may result in transmission to vulnerable patient populations. Since the vaccines do not elicit protective immunity in everyone, many vaccine recipients may want to know how well their vaccine induced protective antibodies, and how long they circulate in peripheral blood. NAb levels have been modeled as correlates of protection from infection and/or disease[8]. Here we report the results of a study in which NAb levels were measured in finger-stick whole blood from mRNA vaccine recipients at 2–4 weeks and 2–4 months after their second dose, and then again pre- and post-3rd mRNA vaccine dose.

Our results demonstrate 25% percent ($n = 67$) of 2-dose vaccine recipients' ($n = 269$) NAb levels show <50% neutralization 2–4 weeks after their second dose and are therefore classified as vaccine poor responders (VPRs). Twenty-three of the 67 VPRs received a third mRNA vaccine as a booster dose. Sixty-five percent of these VPRs received three doses of BNT162b2 (Pfizer), 4% had 3 doses of mRNA-1273 (Moderna), and 30% had 2 doses of BNT162b2 followed by a third dose of mRNA-1273 (booster). Within a month after receiving a third dose, NAb levels in the 23 VPRs increased an average of 5.4-fold, suggesting the importance of a third dose for high levels of peripheral protection.

## Methods

Since performing neutralization assays with authentic SARS-CoV-2 is time-consuming, expensive and requires high-containment facilities with specially trained laboratory personnel, we previously developed a rapid test that semi-quantitatively measures levels of neutralizing antibodies in whole blood or serum. The rapid test utilizes lateral flow technology and is based on the principle that NAbs prevent the receptor binding domain (RBD) on spike protein from binding to ACE2 (Fig. 1)[5,6]. Interpretation of the test is counter-intuitive: the weaker the test line, the stronger the neutralizing activity. Test and control line densities can be quantified with a lateral flow reader and recorded electronically.

### Rapid test to detect SARS-CoV-2 neutralizing antibody

*Study design and population.* Male and female adults ranging in age from 18 to 80 years old were recruited with informed consent to measure their NAb levels using the rapid test after vaccination with either BNT162b2 or mRNA-1273. The study was approved as an observational study by the institutional review board (IRB) at Arizona State University (IRB# 0601000548). In this cohort, no participant ever tested positive by PCR or was diagnosed with COVID-19 prior to the study. NAb levels were measured in all participants 2–4 weeks after a second dose of either BNT162b2 or mRNA-1273 vaccine, then measured 2–4 months after dose 2. In those participants who informed us that they had decided to get a third vaccine dose, NAb levels were measured within 2 weeks of receiving dose 3 of either BNT162b2 or mRNA-1273, and then measured again 2–4 weeks after dose 3.

*Ethical approval.* All data generated in this study used finger-stick peripheral blood collected under an Arizona State University institutional review board (IRB) approved protocol #0601000548. Subjects were assigned a vaccine study de-identification number (VAC-ID) at the time of enrollment and all subsequent collections were conducted in compliance with the Collaborative Institutional Training Program (CITI) Human Subjects Research (HSR) regulatory guidance.

*Assay design and implementation.* The LFA cassette contains a test strip composed of a blood filter overlapped with a conjugate pad, nitrocellulose membrane striped with test and control lines, and an absorbent pad to wick excess moisture. Test strips are secured in a plastic cassette that contains a single sample port. Recombinant ACE2-6xHis protein (Axim Biotechnologies, Inc) is striped onto the nitrocellulose membrane as a test line and an anti-mouse antibody is striped onto nitrocellulose as a control line. Recombinant SARS-CoV-2 Wuhan RBD-6xHis protein (Axim Biotechnologies, Inc) is coupled by carbodiimide chemistry to the surface of 150 nm carboxyl-functionalized gold nanoshells (Nanocomposix Inc). The LFA also contains a control mouse monoclonal antibody (Axim Biotechnologies, Inc) conjugated to the surface of 40 nm carboxyl-functionalized gold nanospheres (Nanocomposix, Inc) and corresponding anti-mouse IgG (Lampire Biological Laboratories) at the control line to ensure the test was performed properly. A mixture of RBD-modified gold nanoshells and a mouse IgG-modified gold nanospheres is dried on the conjugate pad. Linearity of the assay was determined by serial dilutions of strongly neutralizing plasma (reciprocal titer 640-1280) into non-neutralizing (negative) plasma. The assay response is linear up to ~8x dilution of neutralizing serum into non-neutralizing serum. At higher dilutions the test signal approaches levels comparable to a negative sample. Limit of quantitation was adjusted to a reciprocal titer of ~40 using a series of NAb-positive samples with titers assigned by live Wuhan virus FRNT assay. The precision of the test was determined by T-line/C-line ratio to be 4.75%, 8.5% and 15% low (high signal), medium and high (low signal) Nab levels, respectively, with each level run in 20 replicates.

*Determining NAb levels using the rapid test.* To determine NAb titers, 10 μl of finger-stick blood was transferred via micropipette to the sample port in the LFA cassette. After 10–30 s, 60 μl (2 drops) of chase buffer was added to the port. Ten minutes after addition of chase buffer, control and test line densities were

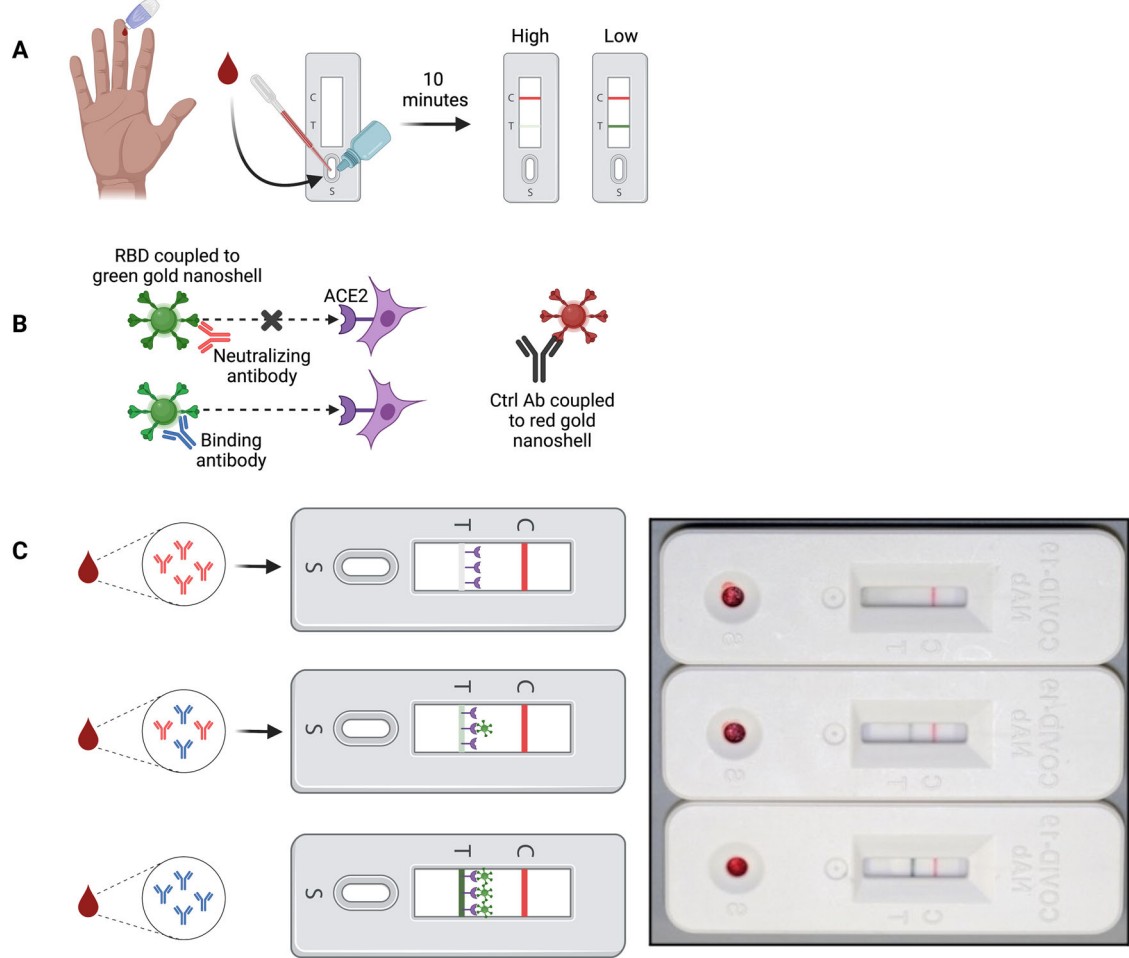

**Fig. 1 Lateral flow assay to detect SARS-CoV-2 neutralizing antibody.** Interpretation of the rapid test is counter intuitive. **A** Methodology overview. Fingerstick blood is transferred to the sample port and followed by two drops chase buffer. Ten minutes later results can be interpreted. Absent or faint test line indicates high NAb levels, while dark or intense test line indicates low/no NAbs. **B** Mechanistic schematic. NAbs bind RBD coupled to a green gold nanoshell (GNS) and *prevent* the RBD/ACE2 interaction from occurring. Abs that bind RBD but do not neutralize *allow* the RBD/ACE2 interaction to occur, shown as increasingly dark signal as more RBD-GNS/ACE2 binds at the test line. **C** Example tests showing highly neutralizing (top), moderately neutralizing (middle), and poorly neutralizing antibodies (bottom) using either finger-stick whole blood. A monoclonal control antibody coupled to a red-GNS runs laterally with the sample/buffer mixture and binds at the control line, seen as a red line. Cartoon image was created by authors using BioRender.com. The picture of three lateral flow cassettes in Fig. 1C is an image from actual test subject samples.

quantified using a Detekt RDS-2500 density reader (Detekt, Austin, TX). Higher titers of NAbs in blood will cause the test line to be weak or absent because they prevent RBD-gold-nanoshells from binding to ACE2 at the test line, while lower titers of NAbs allow RBD-gold nanoshells to bind ACE2 so that a test line is visible. Test line density is inversely proportional to RBD-NAbs present within the sample as previously reported[18].

*Focus reduction neutralization test (FRNT).* To support the application of the rapid test to measure NAb levels to SARS-CoV-2, we correlated LFA test line densities with $IC_{50}$ values obtained in a Focus Reduction Neutralization Test (FRNT) from 38 serum samples[18]. Test performance was evaluated using a correlation regression analysis of $IC_{50}$ values and LFA line densities to obtain the equation, $Y = -0.7698*X + 24.14$ when $X = \log_2 IC_{50}$ as shown in Supplementary Fig. S1 (Supplementary Fig. S1).

**Statistical Analyses.** Levene's test was used to assess homoscedasticity between groups prior to significance testing (IBM SPSS Statistics for Macintosh, Version 26.0; Armonk, NY). To account for unequal variances resulting from unequal sample sizes, Welch's $t$ test with Benjamini-Hochberg false discovery rate (FDR) correction was performed using Microsoft Excel (Version 16.55; Redmond, WA) to evaluate significant differences in mean neutralization between BNT162b2 ($n = 180$) and mRNA-1273 ($n = 89$) 2-4 weeks post-2nd dose. Cohen's $d$ was calculated using Microsoft Excel. Post-hoc power analysis was computed using G*Power 3.1 software[19].

## Results

**Correlation of test line densities to serum $IC_{50}$ values.** To support the application of the LFA to measure NAb levels to SARS-CoV-2, we previously reported[18] correlation of LFA test line densities with $IC_{50}$ values obtained in a Focus Reduction Neutralization Test (FRNT). The rapid test accurately and semi-quantitatively measures levels of NAbs directed against SARS-CoV-2. Serum samples with strong neutralizing activity demonstrate low test line densities while sera with weak neutralizing activity demonstrate high test line densities.

Armed with $IC_{50}$ values, LFA densities and neutralizing serum titers from the FRNT, we calculated % neutralization as: 1-(Test Line Density/Limit of Detection)*100%. Table 1 shows percent

**Table 1 Comparison of LFA density units to IC$_{50}$, NAb titer, and percent neutralization.**

| Image of NAb test result | Test line density unit ranges (thousands) | IC$_{50}$ ranges | Reciprocal NAb titer ranges | % Neutralization ranges |
|---|---|---|---|---|
|  | 10-99 | 17,530 to 880.54 | <1280, ≥640 | 99-90 |
|  | 100-199 | 880.53 to 357.847 | <640, ≥320 | 89-80 |
|  | 200-369 | 357.845 to 160.927 | <320, ≥160 | 79-61 |
|  | 370-599 | 160.927 to 85.88 | <160, ≥80 | 60-36 |
|  | 600-799 | 85.88 to 59.102 | <80, ≥40 | 35-15 |
|  | 800-1000 | 59.101 to 44.23 | ≤40 | ≤15 |

Correlation of IC$_{50}$ values from a Focus Reduction Neutralization Test (FRNT) using authentic SARS-CoV-2 with number of samples per IC$_{50}$ group were re-worded to reflect titer ranges used in the table using five PCR-confirmed samples with IC$_{50}$ values ≤40, five samples with IC$_{50}$ values ≥40 and <80, eight samples with IC$_{50}$ values ≥80 and <160, three samples with IC$_{50}$ values ≥160 and <320, eleven samples with IC$_{50}$ values ≥320 and <640, and six samples with IC$_{50}$ values ≥640. Reciprocal NAb titers were derived using the highest dilution that did not exceed each serum IC$_{50}$ value. Percent neutralization was calculated using the following formula: 1-(Test sample line density/Limit of Detection)*100% where LoD for non-neutralizing sera for the rapid test was 942,481. Limit of detection (LoD) was calculated based on the method of Armbruster and Pry[37], using a convalescent serum sample containing the lowest detectable concentration of analyte (neutralizing antibodies) still distinguishable from a blank. Due to the competitive format of the LFA, the operand was changed to reflect subtraction from limit of blank (LoB) rather than addition: LoD= limit of blank (LoB) − (1.65× SD$_{low conc sample}$): LoD = 1,047,382- (1.65 * 63,769) = 942,481 Test Line Density Units. A lower LoD was not applicable, as polyclonal antisera was used in this study, rather than an individual Mab. Alternatively, the average line density observed for the top 10 donors who demonstrated the strongest ability to prevent RBD from binding to ACE2 was 20,706.

neutralization ranges that correlate to serum titers, FRNT$_{50}$ values and test line densities. Percent neutralization was used throughout the study to measure NAb levels in study participants. Supplementary Figure S2 shows actual LFA tests with density values and corresponding IC$_{50}$ values, NAb titers, and percent neutralization.

**Evaluation of COVID-19 mRNA vaccine NAb response.** NAb levels were measured using our semi-quantitative rapid test in 269 healthy individuals who ranged in age from 19 to 80 (Average age = 51; 165 females and 104 males) who received 2 doses of either BNT162b2 (Pfizer) or mRNA-1273 (Moderna) vaccines[18]. Twenty-three of the 269 were VPRs (neutralized < 50%) and independently received a third dose of either BNT162b2 or mRNA-1273 vaccines. Demographics of the third dose cohort are shown in Table 2.

NAb levels in vaccine recipients were measured at: (i) 2–4 weeks after a second vaccine dose, (ii) 2–4 months after the second dose, (iii) within 1–2 weeks prior to a third dose and (iv) 2–4 weeks after a third mRNA vaccine dose. Several observations were made during this study. Percent neutralization ranged from 0 to 99% 2–4 weeks after a second dose (Fig. 2A). Although our LFA is a surrogate neutralization test, our results agree with previous findings in which the majority (75%) of

vaccine recipients demonstrate NAb levels at ≥50% 2–4 weeks after their second dose[14,15]. Our study also revealed that 25% of vaccine recipients did not neutralize above 50% (Median neutralization = 21%) within a month after their second dose (Fig. 2A).

Twenty-three VPRs ranging in age from 31 to 79 (10 males, 13 females, average age = 62.5, Table 2) independently obtained a third dose of either BNT162b2 or mRNA-1273 vaccine 1-8 months (average = 5 months) after their second dose. Two to four weeks after their third dose, VPRs showed a 5.4-fold increase in NAb levels (range 46%–99%) (Fig. 2B), when comparing average percent neutralization at post-2nd dose and post-3rd dose timepoints, suggesting that most VPRs are not permanently poor responders; they are capable of generating high NAb levels with an additional vaccine dose.

Separating VPRs in Fig. 2A into mRNA-1273 and BNT162b2 vaccine recipients unexpectedly revealed that 14% of mRNA-1273 recipients were VPRs, while 31% of BNT162b2 recipients were VPRs. Only one of twelve mRNA-1273 VPRs chose to receive a third dose of vaccine. In contrast, 23 of 58 BNT162b2 VPRs chose to receive a third dose of either vaccine (see Table 2) as shown in Fig. 3. Statistically, Levene's test indicated heteroscedasticity ($p < 0.001$), while Welch's $t$ test showed significant differences in mean neutralization between groups 2–4 weeks post-2nd dose ($q < 0.001$) with medium effect ($d = 0.537$) and observed power nearing unity ($1–\beta = 0.981$).

## Discussion

Some considerations about our findings include the following. We were surprised to observe that 67/269 (25%) of participants in our study did not demonstrate neutralization >50%. It is not known if poor NAb responders are at increased risk of infection or severe disease. However, anti-viral T cells and antibodies that mediate ADCC are also important components of immunity and prevent disease once a host is infected. Although 50% neutralization corresponds to titers <1:160, it is not known if titers of 1:80, for example, would protect an individual from infection and disease. Likewise, it is not known if individuals with highly neutralizing antibodies corresponding to titers of ≥1:320 would not be protected from infection and disease. However, some models and reports have predicted that NAb levels can serve as a correlate of protection[8,20,21].

The debate about whether a vaccinated individual can transmit virus depends in part on their levels of neutralizing antibodies. NAbs prevent infection and are used therapeutically to treat COVID-19 patients[11]. T cells are crucially important for eliminating infected cells[22–24], but if anti-viral T cells are engaged, the host is already infected. As NAb levels decrease with time after vaccination, there is an increased likelihood that exposure to SARS-CoV-2 could lead to infection which could potentially lead to transmission[25]. This may be an important point since a significant portion of the population has not been vaccinated and could be infected by a vaccinated individual whose NAb levels are low, such that they do not prevent infection and asymptomatically shed virus just prior to reactivation of immune memory.

Twenty-five percent of total participants ($n = 269$) in our study did not generate NAb levels stronger than 50% after a 2-dose regimen. These VPRs ranged in age from 19 to 80 with an average age of 57, median age of 60 ($n = 67$). The age range of non-VPRs was 20–80 with an average age of 50, median age of 51 ($n = 202$). Further studies could be performed to determine the relationship of age and NAb levels <50% after COVID-19 vaccination. Our data suggest that COVID-19 vaccine strategies might follow at least a multiple-dose regimen to keep peripheral NAb levels high, limiting infection, asymptomatic viral replication, and potential

**Table 2 Demographic Information.**

| Age / Sex | RT-PCR results | 1st and 2nd dose vaccine | 3rd dose vaccine | Months post-2nd dose, Prior to 3rd |
|---|---|---|---|---|
| 70-75 / M | Negative^ | BNT162b2 | BNT162b2 | 1 |
| 30-35 / M | Negative* | BNT162b2 | BNT162b2 | 7 |
| 60-65 / M | Negative* | BNT162b2 | BNT162b2 | 8 |
| 50-55 / M | N/A | BNT162b2 | BNT162b2 | 5 |
| 56-60 / F | Negative* | BNT162b2 | BNT162b2 | 6 |
| 60-65 / M | Negative^ | BNT162b2 | BNT162b2 | 7 |
| 50-55 / F | Negative* | BNT162b2 | BNT162b2 | 7 |
| 60-65/ M | N/A | BNT162b2 | BNT162b2 | 6 |
| 66-70 / F | Negative* | BNT162b2 | BNT162b2 | 6 |
| 60-65 / F | Negative^ | BNT162b2 | mRNA-1273 | 1 |
| 56-60/ F | Negative* | BNT162b2 | BNT162b2 | 7 |
| 70-75 / F | Negative~ | BNT162b2 | BNT162b2 | 5 |
| 70-75 / F | N/A | BNT162b2 | BNT162b2 | 6 |
| 76-80 / M | Negative* | BNT162b2 | BNT162b2 | 7 |
| 70-75 / F | Negative* | BNT162b2 | mRNA-1273 | 6 |
| 76-80 / F | N/A | BNT162b2 | mRNA-1273 | 6 |
| 60-65 / F | Negative* | BNT162b2 | mRNA-1273 | 5 |
| 66-70 / M | Negative^ | BNT162b2 | BNT162b2 | 6 |
| 40-45/ M | Negative^ | BNT162b2 | mRNA-1273 | 6 |
| 70-75 / M | Negative^ | mRNA-1273 | mRNA-1273 | 5 |
| 50-55 / F | N/A | BNT162b2 | mRNA-1273 | 3 |
| 70-75 / F | Negative* | BNT162b2 | BNT162b2 | 6 |
| 36-40 / F | Negative* | BNT162b2 | mRNA-1273 | 4 |

^ = TaqPath (Thermo Fisher)
* = Abbott Real Time SARS-CoV-2
~ = PerkinElmer SARS-CoV-2 Real-Time RT-PCR assay
NA = Not Available; participants denied having COVID-19 or being exposed.
Demographic information for 23 study participants who received a 3rd RNA Vaccine Dose. Twenty-two individuals received two doses of BNT162b2 and one individual received two mRNA-1273 doses initially. 15 of the 22 individuals that were originally vaccinated with BNT162b2 obtained a 3rd dose of BNT162b2, and 8 received mRNA-1273 (100 μg) as their 3rd dose. One individual originally vaccinated with mRNA-1273 received a 3rd, 100 μg dose of mRNA-1273. All participants had either confirmed negative RT-PCR results or no known history of infection prior to enrollment. RT-PCR platform indicated using symbols defined below Table S1. Age ranges are provided to protect the identities of the individuals in the study.

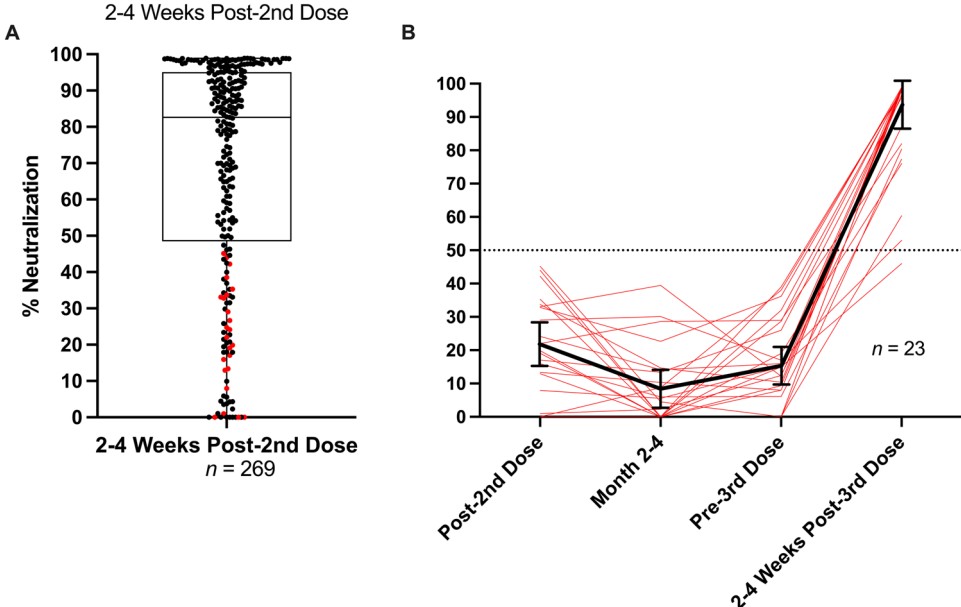

**Fig. 2 NAb profile of mRNA vaccine recipients pre- and post-3rd vaccine dose. A** Spectrum of NAb levels 2–4 weeks post-2nd mRNA vaccine dose (180 BNT162b2 combined with 89 mRNA-1273 vaccine recipients = 269) ranging from 0% neutralization to 99% neutralization. Horizontal line within second and third quartile box denotes median at 83%. Sixty-nine participants in the lower quartile neutralized at <50%. Red dots in lower quartile indicate participants who received 3rd vaccine doses as shown in (**B**). **B** Vaccine Poor Responder Third Dose Recipients Red lines indicate NAb levels in poor responders (<50% neutralization) at 2–4 weeks post second dose, 1–2 weeks prior to a third vaccine dose and 2–4 weeks after a third dose of either BNT162b2 or mRNA-1273. Solid black line is the average % Neutralization of 3rd vaccine dose recipients at each time point with error bars that represent 95% confidence intervals. At 2–4 weeks post-3rd dose the average neutralization was 88%.

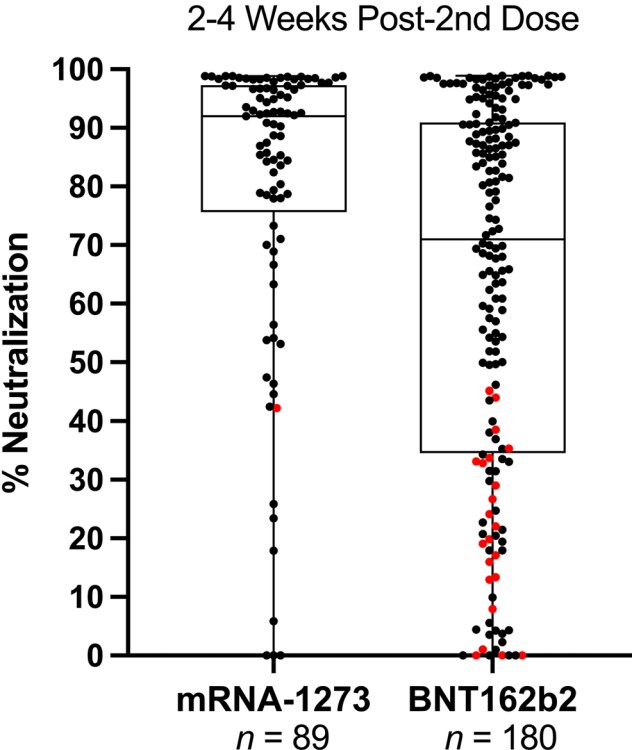

**2-4 Weeks Post-2nd Dose**

**Fig. 3 Comparison of NAbs after 2nd dose of either mRNA-1273 or BNT162b2.** % Neutralization (y-axis) indicates NAb levels ranging from 0 to 99% neutralization. Data shown as box and whisker plots with black vertical lines that denote upper and lower extremes, and horizontal lines that denote upper and lower quartiles with median at the midline. Median neutralization of mRNA-1273 (n = 89) and BNT162b2 (n = 180) is 92% and 71%, respectively. Mean neutralization for mRNA-1273 and BNT162b2 groups is 80% and 63%, respectively. Red dots indicate VPRs that received a 3rd vaccine dose as shown in Fig. 2B, demographics in Table 2.

transmission. It also suggests that NAb levels in vaccine recipients could be evaluated with a rapid test on an individual basis to indicate when an additional dose might be indicated.

Although healthcare policy may recommend that a population should receive a third COVID-19 vaccination at a particular time point, an inexpensive rapid test could provide personalized NAb levels on an individual basis to indicate who might or *might not require* a third dose. Not only would this conserve vaccine, but vaccinating individuals who already have elevated levels of NAbs may not provide benefit since spike protein could be cleared by circulating NAbs as fast as it is made by cells.

Previous reports indicate that NAb levels decline much more rapidly than protection from hospitalization and disease[14,26], but that does not account for vaccine recipients who never generated high levels of NAbs after two doses. Moreover, it is possible that VPRs could be a source of breakthrough infections. Although it is not known what levels of NAbs protect from infection or disease, many vaccine recipients in high-risk professions may wish to keep peripheral NAb levels high, limiting infection, asymptomatic viral replication, and potential transmission.

Although vaccine durability studies indicate an average neutralization geometric mean titer (GMT) of ≥320 during the peak period after 2nd dose[12,27], the distribution among individual serum samples obtained during the observed peak neutralization period (4 to 30 days post-2nd dose) varies greatly[27]. It is unclear what percentage of a population falls below a given GMT or $IC_{50}$

during the peak neutralization period following 2nd dose. Our study supports other findings that majority of healthy individuals generate a NAb response ≥75% neutralization ($IC_{50}$ ≥ 160 and <320). However, we highlight a VPR population that, despite healthy status at the time of vaccination, fail to mount a NAb response >50% ($IC_{50}$ < 160) after two doses.

Poor NAb titers have been reported in special populations such as patients with ongoing cancer therapies[28], solid organ transplant patients[29–31], and individuals on systemic immunosuppressive regimens for various immune-mediated inflammatory diseases[32]. However, current literature is lacking regarding protective antibody responses to COVID-19 in a healthy population. Finally, it is not unprecedented in other vaccine settings such as influenza to observe poor or non-neutralizing responses in healthy individuals[33,34]. Due to the urgency to develop vaccines to slow the COVID-19 pandemic, we are still learning the parameters of mRNA vaccine dose, frequency, timing and durability in the human population.

This study has several limitations. First, it is still unknown what levels of neutralizing antibodies correlate with protection against infection and potential disease. It is possible, but unlikely, that NAb levels as low as 20% could protect against infection[8]. Second, although antibodies directed against the N-terminal domain of spike protein have also been shown to neutralize SARS-CoV-2, it is currently characterized as a minor component of neutralizing antibodies[9,35] and our test does not detect them. Although we measured NAb levels for twice as many BNT162b2 vaccine recipients as mRNA-1273 recipients, we examined homogeneity of variance using Levene's test and, upon confirming unequal variances, assumed Welch's t-test as a conservative and robust alternative to parametric comparisons of means. Importantly, potential for type I error was mitigated using FDR-adjustment of calculated significance, and Cohen's d showed appreciable effect size between groups. Moreover, post-hoc power analysis showed exceptional sensitivity and low chance of type II error, further supporting the significantly lower percent neutralization observed in Pfizer recipients 2–4 weeks post-2nd dose.

Finally, the Omicron variant(s) became widespread since the submission of this manuscript. Although vaccines based on the ancestral Wuhan sequence may not be as effective at preventing infection as an Omicron-based vaccine if it becomes available, additional boosters have been shown in recent publications to be partially effective against virus with Omicron-like mutations and pseudotyped Omicron[36].

In conclusion, our findings suggest that 14% of mRNA-1273 and 31% of BNT162b2 two-dose vaccine recipients ranging in age from 19 to 80 with an average age of 57 (median age of 60) may not have generated levels of NAbs ≥50%, and that additional COVID-19 vaccine doses might be indicated for these individuals. Longitudinal studies are ongoing to determine if high NAb levels in recipients of a third vaccine dose are more durable than NAb levels after two doses.

### Data availability
Source data and the corresponding raw data used to generate Figs. 2, 3, and Supplementary Figure S1 are provided with this manuscript, available in the file 'Supplementary Data'. Source data refers to percent neutralization calculations generated from raw data, as de-identified raw data are recorded in test line density units. The equation used to calculate percent neutralization is described in the Table 1 legend in the main text.

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

## Acknowledgements

We thank Arizona State University School of Life Sciences and Mayo Clinic Arizona Collaborative Research facilities for their providing the laboratory space and time needed to conduct this study at multiple testing location sites.

## Author contributions

D.F.L. and A.J.R. contributed equally to this work as joint first authors, writing the manuscript, generating and editing figures. S.S., M.G.M., and A.S.-N. contributed to development of the NAb LFA and supplied test materials. A.J.R. and M.A.K. conducted data analysis and collection. M.A.K., J.V., D.M., J.F., B.E. helped with data collection and recording. E.J.K. provided insight to study design. P.J. contributed to statistical analysis.

## Competing interests

Co-authors A.J.R., M.A.K., J.V., D.M., J.F., B.E., E.J.K., and P.J. declare no competing interests. D.F.L., S.S., A.S.-N., and M.G.M. declare the existence of a financial competing interest. D.F.L. and S.S. are co-founders of Sapphire, the research division of AXIM Biotechnologies. S.S., M.G.M,. and A.S.-N. are employed by AXIM. All other authors have no competing interests and no financial relationships with any organizations that might have an interest in the submitted work in the previous three years, no other relationships or activities that could appear to have influenced the submitted work.
