## [Peer Review File · Communications Medicine]

Reviewers' comments:

Reviewer #2 (Remarks to the Author):

The authors used a newly developed lateral flow, ACE-2 binding-inhibition assay (LFA) to measure SARS-CoV-2 neutralizing antibodies in a small cohort of mRNA-vaccinated subjects who mounted a weak or undetectable neutralizing antibody response after two doses of the vaccines. They provide preliminary evidence showing these individuals develop a stronger neutralizing antibody response after a boost. They conclude that “vaccine poor responders” are not permanently poor responders because they make high NAb levels after an additional vaccine dose.

While these findings are potentially interesting, the manuscript has as a number of weaknesses:

- 1) The LFA is poorly described (e.g., no mention of specific reagents used, how reagents were coupled and processed into as assay) and has a notable lack of sensitivity, detecting positive neutralization in only 75% of mRNA vaccine recipients, whereas most neutralization assays detect positive activity in nearly 100% of people who receive these vaccines. There is also a lack of information on the linearity, precision and limits of quantification of the assay. These are very important parameters to define for a new assay that is being used as a surrogate for neutralization.
- 2) ID50 neutralization titers in the LFA are inferred from a correlation seen with a live virus assay using a separate set of samples. Is the accuracy of the live virus assay known?
- 3) The main dataset in Fig 2 does not illustrate a 20-fold increase in NAb levels as described in the text. Is this 20-fold increase inferred from the correlation analysis with live virus assay results?
- 4) Lines 87-89: The authors attempt to describe antibody effector functions that may contribute to protection in addition to neutralizing antibodies. These effector functions are more broad and complex than indicated.
- 5) Second para of Intro: The authors place a heavy emphasis on protection against asymptomatic infection, which has become very difficult to achieve with the omicron variant. They seem to be implying that lack of protection against asymptomatic infection and unknowingly having potential to transmit, especially for health care workers, should be a policy. Although this makes intuitive sense, the existing data for omicron suggest this policy would be futile.
- 6) Lines 103-104: The authors should be made aware of recent publications showing neutralizing antibodies correlated with protection in COVID-19 efficacy trials (<https://doi.org/10.1038/s41591-021-01540-1>; <https://www.science.org/doi/10.1126/science.abm3425>).
- 7) Line 112: Clarify that they received a boost rather than 3 additional doses of the vaccines.
- 8) Table 2: BNT162b2 and mRNA-1273 are the names of the vaccines, not the manufacturers.
- 9) The discussion is again heavily focused on protection from infection when it would be more appropriate to focus on protection from severe disease. This is a major weakness of the rationale for this study. The reader gets the impression that NAb's are only effective if they prevent infection, which may very well be wrong for SARS-CoV-2.

10) Lines 358-359: "Previous reports indicate that NAb levels decline much more rapidly than protection from hospitalization and disease (12,22)." This is a confusing statement because the protective levels of NAbs are not known.

11) Fig S2 needs a sharper image.

Reviewer #3 (Remarks to the Author):

This lateral flow neutralizing Ab (NAb) test reported here is a step in the right direction. There are self administered rapid lateral flow tests now in widespread use to detect antigens in nasal and throat swabs. There are no rapid tests for detecting whether a person has neutralizing Abs against COVID.

There is an EUA approved neutralizing Ab test, a 96-well type ELSIA assay that can be run in clinical diagnostic laboratories. It uses a similar assay principle as reported here, measuring Ab mediated blocking of RBA-ACE2 binding, but it should be run in a CLIA laboratory is not a rapid lateral flow test that can be self-administered.

As with the EUA approved test, this is a surrogate of real neutralization. There is no NAb titer from this assay that will predict that a person will not come down with COVID.

The authors should compare their results with the EUA approved ELISA, which uses a very similar assay principle.

The r-value between their assay and the gold standard is OK in recovered convalescent patients is just OK. They should also compare virus neutralizing activity and RBD/ACE2 blocking activity in vaccinees. The observed correlations may not be as good when vaccinees virus neutralizing titers and RBD-ACE2 binding are compared. This is important because one of the most important assay will be to show NAb in the Vaccinees not just in the convalescent population.

Reviewer #4 (Remarks to the Author):

The work describes the use of a lateral flow test to measure SARS-CoV-2 spike RBD-binding antibodies in vaccinated individuals, following either 2 or 3 doses of an mRNA vaccine. The assay is quite novel and has great utility in being able to quickly measure anti-spike RBD antibodies directly from a finger-prick bleed. In the future this could allow healthcare practitioners to be able to make an on-the-spot decision as to whether to give a booster shot to a patient- although further data regarding what constitutes a protective threshold of NAbs is also required. Using this assay the authors find that a substantial percentage of individuals vaccinated with 2 doses of mRNA vaccine do not reach the 50% neutralisation target set in their assay (<1:160 dilution via a standard FRNT), however a third mRNA vaccine dose increased this response in most individuals. However, how this compares to antibodies responses in individuals who were initially 'good responders' following 2 doses is unknown.

Major points:

1. It is shown that NABs in almost all VPRs increase following a 3rd dose of an mRNA vaccine. However, there is no comparison shown with 'good responders'. Therefore, it is unknown if the response in VPRs is still well below that of the good responders following a 3rd vaccine dose. Although the VPRs reach a high level of neutralisation in this particular assay following dose 3, the good responders may still exceed this maximum level, and so the conclusion 'that poor responders are not always poor responders' is not valid.
2. Figure 3 compares the number of VPRs who received mRNA-1273 vs. BNT162b2 and notes that the later group has a higher percentage of VPRs. Although this is not unexpected given the larger mRNA dose given with mRNA-1273, without including the underlying health status and age of the recipients in each group, it is hard to know if these factors may have biased the results.
3. According to last column of table 3, some of these VPRs received a 3rd dose 1-3 months post second dose-is this correct and is this because they were expected to be poor responders? Why such a short interval if they were healthy individuals?

Minor points:

1. Line 82 should be re-worded. Vaccines prevent serious disease in 95% of individuals as states, however this does not mean that these 95% of individuals are protected from infection.
2. Lines 104-106 are duplicated in lines 108-110.
3. Line 112 has the first use of "VPR" in main body of text, should be spelt out in full here.
4. Lines 112-113 should be re-worded, currently it implies that individuals were given 3 doses on top of already receiving the initial 2 doses of vaccine.
5. Line 251-252 It has been shown that there can be substantial inter-lab variability in neutralisation titres (e.g. Cromer et al. [https://doi.org/10.1016/S2666-5247\(21\)00267-6](https://doi.org/10.1016/S2666-5247(21)00267-6)) and given this is a unique assay, I don't think it can be directly compared to NABs reported in the other studies i.e. references 12 and 13.
6. The vaccines used and antibodies measured are all based on the ancestral spike protein, the impact of variants, e.g. Omicron, that exhibit some ability to evade NABs, and the importance of boosters in this context, should be discussed.

March 21, 2022

Dear Communications Medicine Editors;

We thank the reviewers for their comments and thoughtful review. In blue text we have responded to each of the reviewer's concerns, provided some "reviewers eyes only" data that addresses some questions raised and indicated in the responses where changes were made in the main text. We hope reviewers will find this revised manuscript acceptable.

Reviewers' comments:

Note: Reviewer #1 did not provide a review.

Reviewer #2 (Remarks to the Author):

The authors used a newly developed lateral flow, ACE-2 binding-inhibition assay (LFA) to measure SARS-CoV-2 neutralizing antibodies in a small cohort of mRNA-vaccinated subjects who mounted a weak or undetectable neutralizing antibody response after two doses of the vaccines. They provide preliminary evidence showing these individuals develop a stronger neutralizing antibody response after a boost. They conclude that "vaccine poor responders" are not permanently poor responders because they make high NAb levels after an additional vaccine dose.

While these findings are potentially interesting, the manuscript has as a number of weaknesses:

1) The LFA is poorly described (e.g., no mention of specific reagents used, how reagents were coupled and processed into as assay) and has a notable lack of sensitivity, detecting positive neutralization in only 75% of mRNA vaccine recipients, whereas most neutralization assays detect positive activity in nearly 100% of people who receive these vaccines. There is also a lack of information on the linearity, precision and limits of quantification of the assay. These are very important parameters to define for a new assay that is being used as a surrogate for neutralization.

We wish to clarify Reviewer #2's mis-interpretation of the main idea of our findings. After 2 doses of vaccine, 75% of vaccine recipients demonstrated $\geq 50\%$ neutralization in our semi-quantitative rapid test such that 50% neutralization correlates to a serum titer of $\geq 1:160$ as shown in Table 1. Accordingly, 25% of "2-dose mRNA vaccine recipients" neutralized SARS-CoV-2 at $< 50\%$. We wish to clarify that these vaccine recipients were detectable semi-quantitatively by our assay, but failed to surpass 50% neutralization. However, as detailed in our previous publication, the LFA values and calculated neutralization values obtained from these "poorly neutralizing" individuals are distinguishable from individuals that were neither vaccinated nor infected. The LFA was thoroughly described in a previously published paper: Lake DF et al. Development of a rapid point of care test that measures neutralizing antibodies to SARS-CoV-2. *J Clin Virol.* 2021.145:105024. In that publication, we detailed precision and linearity between highly neutralizing serum samples and poorly neutralizing samples. For "reviewer's eyes only" we have copied below the appropriate methods section from our *J Clin Virol* paper. In the current manuscript, Table 1 is an important "Rosetta stone" in which LFA line density is translated to IC50 (for virologists), titers (for clinicians) and % neutralization (for anyone).

Lateral Flow Neutralizing Antibody Assay

The Lateral Flow NAb assay was developed to measure levels of antibodies that compete with ACE2 for binding to RBD. The LFA single port cassette (Empowered Diagnostics) contains a test strip composed of a sample pad, blood filter, conjugate pad, nitrocellulose membrane striped with test and control lines, and an absorbent pad (Axim Biotechnologies Inc). The LFA also contains a control mouse antibody conjugated to red gold nanospheres and corresponding anti-mouse IgG striped at the control line.

LFAs were run at room temperature on a flat surface for 10 minutes prior to reading results. To perform the test, 6.7µl of serum or 10ul whole blood were added to the sample port followed by 60µl of chase buffer. After 10 minutes, densities of both test and control lines were recorded in an iDetekt RDS-2500 density reader.

The test leverages the interaction between RBD-conjugated green-gold nanoshells (Nanocomposix) that bind ACE2 at the test line when RBD-neutralizing antibodies (RBD-NAbs) are absent or low. Test line density is inversely proportional to RBD-NAbs present within the sample. As a semi-quantitative test, the results of the LFA can be interpreted using a scorecard or a densitometer. A red line across from the “C” indicates that the test ran properly. An absent or faint test line indicates high levels of RBD-NAbs, whereas a dark test line suggests low or lack of RBD-NAbs.

Precision testing was performed using sera from one highly, and one non-neutralizing donor in replicates of 10. Density values were recorded as above and %CVs calculated using the formula: (Standard Deviation/Mean) * 100%.

2) ID50 neutralization titers in the LFA are inferred from a correlation seen with a live virus assay using a separate set of samples. Is the accuracy of the live virus assay known?

The live virus assay used to evaluate LFA correlation refers to a viral focus-forming assay (FFA), also referred to as focus-reduction neutralization test (FRNT) or microneutralization assay (MNA). This neutralization assay is a microtiter well version of a plaque reduction neutralization test (PRNT) which is considered a gold standard for determining serum neutralization titers. Our live virus assays were performed according to FFA/FRNT methods described by Diamond M et al (<https://doi.org/10.1016/j.virol.2020.05.015>). While the accuracy is not defined in their methods, Bewley et al. (<https://doi.org/10.1038/s41596-021-00536-y>) evaluated performance of the MNA using similar methodology, compared to gold standard PRNT. MNA and PRNT are highly correlated (Pearson $r = 0.963$; $P < 0.001$). While also not specifically defined by the authors, a high accuracy can be inferred from the correlation shown against the gold standard (<https://www.nature.com/articles/s41596-021-00536-y/figures/10>). Due to four-fold higher throughput, decreased incubation time, and strong correlation with traditional PRNT, the FFA/FRNT/MNA has become widely used for characterization of SARS-CoV-2 neutralizing antibody activity. Additionally, FDA considers FRNT an equivalent to PRNT.

3) The main dataset in Fig 2 does not illustrate a 20-fold increase in NAb levels as described in the text. Is this 20-fold increase inferred from the correlation analysis with live virus assay results?

Although the text states “average” fold increase, it should be clarified that this value refers to the average of paired sample % neutralization fold-increase for the 23 individuals shown in Fig 2. A **20-fold** increase as described in the text refers to the average of fold-increases for each individual, rather than the fold increase of the average of all individuals.

Average of fold increases:

Fold-increase was calculated for each individual according to their pre-3rd and post-3rd dose % neutralization value, *then* averaged. For example, VAC-37 was at 1% and 60% pre- and post-3rd dose, therefore had 60-fold increase. However, VAC-238 demonstrated only a 2.7-fold increase as they were at 36% pre-3rd and 99% neutralization post-3rd dose. For n=23, the fold increase in this context was **19.72**.

Fold increase of averages:

For the 23 individuals, average % neut pre-3rd and post-3rd dose (n=23) was 16% and 88%, respectively. The fold increase in this context would be **5.36**. We have changed the text in the Results section of the abstract and in lines 282-285 in the Results section of the main text.

4) Lines 87-89: The authors attempt to describe antibody effector functions that may contribute to protection in addition to neutralizing antibodies. These effector functions are more broad and complex than indicated.

We completely agree that antibody effector functions are more broad and complex than indicated in the brief introduction section of our manuscript. We did not intend to elaborate on effector functions of antibodies. We are focused on the importance of SARS-CoV-2 neutralizing antibodies. We have softened the statement on lines 88-90 to: Except for antibodies that mediate antibody-dependent cellular cytotoxicity and complement dependent cytotoxicity, ~~the only~~ antibodies of primary importance are neutralizing antibodies (NAbs).

5) Second para of Intro: The authors place a heavy emphasis on protection against asymptomatic infection, which has become very difficult to achieve with the omicron variant. They seem to be implying that lack of protection against asymptomatic infection and unknowingly having potential to transmit, especially for health care workers, should be a policy. Although this makes intuitive sense, the existing data for omicron suggest this policy would be futile.

We agree that protection against asymptomatic infection has become very difficult to achieve with the omicron variant. The manuscript was written prior to the emergence of omicron and reviewed after omicron became widespread. We wish to clarify that our

intention is to report our observations of levels of neutralizing antibodies in a study of 2nd and 3rd dose vaccine recipients, not to dictate policy.

6) Lines 103-104: The authors should be made aware of recent publications showing neutralizing antibodies correlated with protection in COVID-19 efficacy trials (<https://doi.org/10.1038/s41591-021-01540-1>; <https://www.science.org/doi/10.1126/science.abm3425>).

Thank you for highlighting the importance of these publications in Science and Nature Medicine. We have added the references at the end of the first paragraph in the discussion. We are aware of these and other publications that support neutralizing antibodies as a correlate of protection, however, a threshold of protection remains undefined.

7) Line 112: Clarify that they received a boost rather than 3 additional doses of the vaccines.

The text has been revised to clarify that a “third dose” is a “booster” dose on line 112.

8) Table 2: BNT162b2 and mRNA-1273 are the names of the vaccines, not the manufacturers.

Table 2 column heading revised.

9) The discussion is again heavily focused on protection from infection when it would be more appropriate to focus on protection from severe disease. This is a major weakness of the rationale for this study. The reader gets the impression that NABs are only effective if they prevent infection, which may very well be wrong for SARS-CoV-2.

This is an ideological argument. NABs protect the host from severe disease and hospitalization (the goal of all vaccines and Mab therapies) which is crucial from a public health perspective. We understand this it is not realistic to completely prevent infection. However, many scientists believe that limiting infection is the best way to control viral replication which results in mutations that may escape immune control (both B cells and T cells). We addressed this issue in the discussion in lines 365-373, to include protection from infection and severe disease.

10) Lines 358-359: “Previous reports indicate that NAB levels decline much more rapidly than protection from hospitalization and disease (12,22).” This is a confusing statement because the protective levels of NABs are not known.

We wish to highlight the reports from Widge *et al.* (Ref 12) and Thomas *et al.* (Ref 22) that indicate that protection from hospitalization and severe disease is more durable than neutralizing antibody levels. T cells are likely important preventing hospitalization and severe disease.

11) Fig S2 needs a sharper image.

The images shown in Fig S2 were taken by the DETEKT 2500 densitometer under controlled light settings, such that all tests are imaged under the same conditions. The image quality cannot be changed as it is pre-defined by the densitometer, however, the image has been sharpened as requested.

Reviewer #3 (Remarks to the Author):

This lateral flow neutralizing Ab (NAb) test reported here is a step in the right direction. There are self administered rapid lateral flow tests now in widespread use to detect antigens in nasal and throat swabs. There are no rapid tests for detecting whether a person has neutralizing Abs against COVID.

There is an EUA approved neutralizing Ab test, a 96-well type ELSIA assay that can be run in clinical diagnostic laboratories. It uses a similar assay principle as reported here, measuring Ab mediated blocking of RBA-ACE2 binding, but it should be run in a CLIA laboratory is not a rapid lateral flow test that can be self-administered.

As with the EUA approved test, this is a surrogate of real neutralization. There is no NAb titer from this assay that will predict that a person will not come down with COVID. The authors should compare their results with the EUA approved ELISA, which uses a very similar assay principle.

The r-value between their assay and the gold standard is OK in recovered convalescent patients is just OK. They should also compare virus neutralizing activity and RBD/ACE2 blocking activity in vaccinees. The observed correlations may not be as good when vaccinees virus neutralizing titers and RBD-ACE2 binding are compared. This is important because one of the most important assay will be to show NAb in the Vaccinees not just in the convalescent population.

We benchmarked our LFA using 36 serum samples with known IC50 values and known titers in a gold-standard FRNT in our previously published JCV paper. This test allowed us to directly correlate test line densities with serum IC50 values obtained in a FRNT assay in a BSL3 laboratory. For this reason, comparison of our LFA values against the EUA approved ELISA is less relevant than our comparison to gold-standard.

The reviewer requested comparison of virus neutralizing activity and RBD-ACE2 blocking activity in vaccinees. We went back into the BSL3 laboratory and performed FRNTs on 12 serum samples from previously vaccinated individuals. The r-value using convalescent sera in Supplementary Figure S1 in our previous submission was -0.7918 with $p < 0.001$. The new r-value for vaccinated individuals is -0.7747 with $p < 0.001$, which is nearly identical to previous r- and p-values using convalescent sera.

Reviewer #4 (Remarks to the Author):

The work describes the use of a lateral flow test to measure SARS-CoV-2 spike RBD-binding antibodies in vaccinated individuals, following either 2 or 3 doses of an mRNA vaccine. The assay is quite novel and has great utility in being able to quickly measure anti-spike RBD antibodies directly from a finger-prick bleed. In the future this could allow healthcare practitioners to be able to make an on-the-spot decision as to whether to give a booster shot to a patient- although further data regarding what constitutes a protective threshold of NAb is also required. Using this assay the authors find that a substantial percentage of individuals vaccinated with 2 doses of mRNA vaccine do not reach the 50% neutralisation target set in their assay (<1:160 dilution via a standard FRNT), however a third mRNA vaccine dose increased this response in most individuals. However, how this compares to antibodies responses in individuals who were initially ‘good responders’ following 2 doses is unknown.

For Reviewer eyes only, shown below is a graph of “good responders” (vaccine recipients who neutralized at $\geq 50\%$). An updated version of this graph will be presented in a 2nd dose vs. 3rd dose durability study that is concluding soon. After a 3rd dose, “Good Responders” average 13% better than poor responders.

Reviewer eyes only Figure. The average of NAb levels in “Good Responders” ($\geq 50\%$ neutralization) 2-4 weeks after second vaccine dose is 83%. The average NAb level 2-4 months after a second dose of either RNA vaccine is 40%. The average of NAb levels 1-2 weeks prior to a 3rd dose was 26% and the average NAb level 2-4 weeks after a 3rd dose of either RNA vaccine was 96%.

Major points:

1. It is shown that NAb in almost all VPRs increase following a 3rd dose of an mRNA vaccine. However, there is no comparison shown with 'good responders'. Therefore, it is unknown if the response in VPRs is still well below that of the good responders following a 3rd vaccine dose. Although the VPRs reach a high level of neutralisation in this particular assay following dose 3, the good responders may still exceed this maximum level, and so the conclusion 'that poor responders are not always poor responders' is not valid.

See graph above.

2. Figure 3 compares the number of VPRs who received mRNA-1273 vs. BNT162b2 and notes that the later group has a higher percentage of VPRs. Although this is not unexpected given the larger mRNA dose given with mRNA-1273, without including the underlying health status and age of the recipients in each group, it is hard to know if these factors may have biased the results.

We did not discriminate based on health status, ethnicity, gender or age (between 18 and 80). As such we did not collect data from participants concerning co-morbidities and underlying health status. However, all participants were between the ages of 18 and 80 and were ambulatory and in good general health (no underlying malignancy or immunosuppression).

3. According to last column of table 3, some of these VPRs received a 3rd dose 1-3 months post second dose-is this correct and is this because they were expected to be poor responders? Why such a short interval if they were healthy individuals?

During data collection for the study, there was no federal guidance for a third vaccine dose (booster). As a result some participants decided to obtain a third vaccine dose (booster) independently on their own timing.

Minor points:

1. Line 82 should be re-worded. Vaccines prevent serious disease in 95% of individuals as states, however this does not mean that these 95% of individuals are protected from infection.

The second sentence in the introduction has been re-worded to indicate that 5% of vaccine recipients are potentially susceptible to severe disease.

2. Lines 104-106 are duplicated in lines 108-110.

We deleted lines 108-110 and revised introduction paragraph 3.

3. Line 112 has the first use of “VPR” in main body of text, should be spelt out in full here.

Revised to include VPR definition at first use in main body text.

4. Lines 112-113 should be re-worded, currently it implies that individuals were given 3 doses on top of already receiving the initial 2 doses of vaccine.

Revised introduction paragraph 3 to clarify booster doses were received, rather than additional 3 doses after the initial 2 doses.

5. Line 251-252 It has been shown that there can be substantial inter-lab variability in neutralisation titres (e.g. Cromer et al. [https://doi.org/10.1016/S2666-5247\(21\)00267-6](https://doi.org/10.1016/S2666-5247(21)00267-6)) and given this is a unique assay, I don't think it can be directly compared to NABs reported in the other studies i.e. references 12 and 13.

The reviewer is certainly correct that there can be substantial inter-lab variability in neutralization titres, but we benchmarked the semi-quantitative lateral flow assay reported here and in Lake DF et al. *J Clin Virol.* 2021.145:105024 with IC₅₀ values obtained in a FRNT using the same serum samples (Supplemental Figure S2).

6. The vaccines used and antibodies measured are all based on the ancestral spike protein, the impact of variants, e.g. Omicron, that exhibit some ability to evade NABs, and the importance of boosters in this context, should be discussed.

We agree with the reviewer that Omicron evades NABs induced by Wuhan-based vaccines and have included a discussion in lines 434-438 about boosters enhancing neutralizing capacity against Omicron.

Reviewers' comments:

Reviewer #2 (Remarks to the Author):

Disappointingly, this revised manuscript does little to address some of my concerns:

1. There is still no mention of the specific reagents used, how the reagents were coupled and processed into an assay. The authors claim these details are found in a previous manuscript when in fact they are not. The authors also fail to mention what SARS-CoV-2 variant RBD was used in the LFA and live virus FRNT neutralization assays, or whether the same variant was used in both assays.
2. The authors claim that linearity, precision and limits of quantification of the assay were defined in a previous publication. All I could find in their previous publication was a small experiment that assessed intra-assay repeatability.
3. I am not aware that any neutralization assay for SARS-CoV-2 is considered a gold standard. To date the FDA has relied most on pseudovirus neutralization assays for regulatory decisions.
4. The authors did not address my concern about calling out ADCC and complement dependent cytotoxicity without placing these activities in a broader context of antibody effector functions that could impact the virus. As a result, the authors give the appearance that these are the only two effector functions thought to impact SARS-CoV-2. In fact, I know of no reports implicating complement mediate cytotoxicity.
5. The authors completely side-stepped my request to clarify the distinction and importance of differentiating between protection against asymptomatic infection and protection from disease.
6. The authors say they added references on nAbs as a correlate of protection against symptomatic infection with SARS-Cov-2 but I did not see them in the revised manuscript I downloaded from the reviewer's portal.

Despite these failings, the observation that people who are poor responders after 2 doses of an mRNA vaccine are not inherently destined to remain poor responders and can actually generate high titers of neutralizing antibodies after a boost is important new information.

Reviewer #3 (Remarks to the Author):

Lake and colleagues have described the use of a lateral flow assay to assess Nab titers in a cohort of individuals. They show the efficacy of the third vaccine, or "booster" has on rescuing low antibody titers in individuals who are deemed poor responders. Although there is more work to be done in the field, to assess the long-term effects of the booster, this is a step in the right direction. The authors have appropriately addressed my comments and I endorse publication of this manuscript in Nature Communications Medicine.

Reviewer #4 (Remarks to the Author):

I am satisfied with the amendments to the manuscript, however I note that line 394 of the discussion needs correcting to state that "antibodies directed against the N-terminal domain of

spike...etc"

We wish to thank the reviewers for their thoughtful critiques.

Reviewer #2 (Remarks to the Author):

Disappointingly, this revised manuscript does little to address some of my concerns:

Critique: 1. There is still no mention of the specific reagents used, how the reagents were coupled and processed into an assay. The authors claim these details are found in a previous manuscript when in fact they are not. The authors also fail to mention what SARS-CoV-2 variant RBD was used in the LFA and live virus FRNT neutralization assays, or whether the same variant was used in both assays.

2. The authors claim that linearity, precision and limits of quantification of the assay were defined in a previous publication. All I could find in their previous publication was a small experiment that assessed intra-assay repeatability.

Response: The RBD coupled to gold beads used in this report is based on the original Wuhan sequence. Additionally, FRNTs were performed with the original Wuhan virus, verified by nucleic acid sequence. We included an additional methods section within lines 160-180, "Assay Design and Implementation". This section describes details about the rapid test that the reviewer is requesting.

3. I am not aware that any neutralization assay for SARS-CoV-2 is considered a gold standard. To date the FDA has relied most on pseudovirus neutralization assays for regulatory decisions.

Response: We submitted an EUA for the rapid test used in this study using a template provided by the FDA (serology_neutralization_template.docx). The FDA does NOT allow pseudovirus assays as a comparator method. The following is a quote from the FDA template: **"At this time, Plaque Reduction Neutralization Test (PRNT) is considered the gold standard for detection and measurement of neutralizing antibody titers. Microneutralization assays and Focus Reduction Neutralization Testing (FRNT) are also considered appropriate neutralization comparator methods if no fluorescence is measured."**

4. The authors did not address my concern about calling out ADCC and complement dependent cytotoxicity without placing these activities in a broader context of antibody effector functions that could impact the virus. As a results, the authors give the appearance that these are the only two effector functions thought to impact SARS-CoV-2. In fact, I know of no reports implicating complement mediate cytotoxicity.

Response: We acknowledge that broad antibody effector functions impact disease course after infection with SARS-CoV-2. In addition to SARS-CoV-2, ADCC has been shown as a cytotoxicity mechanism in other viral infections such as HIV and measles. We do not know of any report indicating that complement dependent cytotoxicity occurs against SARS-CoV-2, but it is possible in theory. We are reporting on neutralizing antibodies after vaccination and do not wish to discuss antibody effector molecules or cells, as they are not the focus of the manuscript. In the previous version we wanted to mention that they exist and play an important role in immune responses to infection. However, we decided to remove any mention of ADCC and complement from the introduction in lines 87-90.

5. The authors completely side-stepped my request to clarify the distinction and importance of differentiating between protection against asymptomatic infection and protection from disease.

Response: We apologize for the lack of clarity and have revised the text in introduction lines 98-104 to reflect the importance of differentiating disease from infection. We wish to highlight that at this time, a NAb threshold for use as a correlate of protection has yet to be defined and remains a complex, evolving issue. To specifically address the reviewer's concern we distinguished that a SARS-CoV-2 infection in vaccinated "healthy" individuals may result in disease in vulnerable (immunosuppressed or elderly) populations. We thank the reviewer for their comment and hope they find this revision satisfactory.

6. The authors say they added references on nAbs as a correlate of protection against symptomatic infection with SARS-Cov-2 but I did not see them in the revised manuscript I downloaded from the reviewer's portal.

Response: Although we included references on NAbs as a correlate of protection (Khoury DS et al., and Addetia A, et al.), we neglected to add the reference the reviewer suggested. It is now included as reference # 21. We apologize for the omission.

Despite these failings, the observation that people who are poor responders after 2 doses of an mRNA vaccine are not inherently destined to remain poor responders and can actually generate high titers of neutralizing antibodies after a boost is important new information.

Reviewer #3 (Remarks to the Author):

Lake and colleagues have described the use of a lateral flow assay to assess Nab

titers in a cohort of individuals. They show the efficacy of the third vaccine, or "booster" has on rescuing low antibody titers in individuals who are deemed poor responders. Although there is more work to be done in the field, to assess the long-term effects of the booster, this is a step in the right direction. The authors have appropriately addressed my comments and I endorse publication of this manuscript in Nature Communications Medicine.

Reviewer #4 (Remarks to the Author):

I am satisfied with the amendments to the manuscript, however I note that line 394 of the discussion needs correcting to state that "antibodies directed against the N-terminal domain of spike...etc"

Response: This statement has been corrected. Thank you for finding the error.

REVIEWERS' COMMENTS:

Reviewer #2 (Remarks to the Author):

The authors have now adequately addressed my remaining concerns.